# The Growth of Hexagonal Boron Nitride Quantum Dots on Polycrystalline Nickel Films by Plasma-Assisted Molecular Beam Epitaxy

Nurzal Nurzal [1,2], Wei-Cyuan Huang [1], Cheng-Yu Liu [1], Su-Hua Chen [1] and Ing-Song Yu [1,*]

[1] Department of Materials Science and Engineering, National Dong Hwa University, Hualien 97401, Taiwan; nurzall@gmail.com (N.N.); pioneer840403@gmail.com (W.-C.H.); 810722101@gms.ndhu.edu.tw (C.-Y.L.); shchen@gms.ndhu.edu.tw (S.-H.C.)

[2] Department of Mechanical Engineering, Institut Teknologi Padang, Padang 25143, Kp Olo, Indonesia

* Correspondence: isyu@gms.ndhu.edu.tw; Tel.: +886-3-8903219

**Abstract:** In this report, quantum dots of hexagonal boron nitride (h-BN) were fabricated on the surface of polycrystalline Ni film at low growth temperatures (700, 750, and 800 °C) by plasma-assisted molecular beam epitaxy. Reflection high-energy electron diffraction could trace the surface condition during the growth and perform the formation of BN. The observation of surface morphology by scanning electron microscopy and atomic force microscopy showed the nanodots of BN on Ni films. The existence of crystal h-BN quantum dots was determined by the analysis of Raman spectra and Kevin probe force microscopy. The cathodoluminescence of h-BN quantum dots performed at the wavelength of 546 and 610 nm, attributed to the trapping centers involving impurities and vacancies. Moreover, the influence of temperatures for the substrate and boron source cell was also investigated in the report. When the k-cell temperature of boron and growth temperature of substrate increased, the emission intensity of cathodoluminescence spectra increased, indicating the better growth parameters for h-BN quantum dots.

**Keywords:** hexagonal boron nitride; quantum dots; molecular beam epitaxy; cathodoluminescence; kelvin probe force microscopy



## 1. Introduction

Boron nitride (BN) is a chemically stable material in the group III-V compounds, applied especially for electronic and optoelectronic devices [1,2]. Polymorphism of BN crystal structures include hexagonal (h-BN), rhombohedral (r-BN), turbostratic (t-BN), wurtzite (w-BN), and cubic (c-BN). h-BN has a similar structure to graphene with 1.7% lattice mismatch, as a two-dimensional (2D) material [3]. Excellent physical properties of h-BN include high thermal conductivity [4] and wide bandgap [5]. h-BN is not only a 2D material applied for electronic devices as an insulating layer or a quantum tunnel barrier, but it also has high potential for the applications in deep-ultraviolet light-emitting diodes (LEDs) [6].

For the fabrication of h-BN, several techniques have been employed to grow and produce h-BN. Ion beam sputtering deposition (IBSD) [7], metal organic chemical vapor deposition (MOCVD) [8], and plasma-assisted molecular beam epitaxy (PA-MBE) can be used for the growth of h-BN thin films [9–11]. Among them, PA-MBE has an exact growth control to produce high-quality epitaxy and at lower growth temperatures than others [12,13]. h-BN has also attracted attention for the growth on various substrates, such as Ni [10,14], graphene [15,16], cobalt [17,18], and sapphire [19,20].

Over the past few decades, semiconductor materials have been produced in low-dimensional nanostructures as quantum wells (2D), wires (1D), and dots (0D) to obtain novel devices by exploiting their quantum confinement effect. Quantum dots (QDs) are

free-standing nanoparticles with nanometers. QDs have different optical and electronic properties from larger particles due to quantum mechanics [21]. QDs have unique properties for applications in batteries [22], LEDs [23], biosensors [24], and cancer therapy [25]. QDs of h-BN have a wide bandgap and excellent chemical stability at high temperatures. However, the study about h-BN quantum dots is infrequent so far.

In this work, we focus on the growth of h-BN QDs on the polycrystalline Ni substrates at a relatively low temperature by PA-MBE system. The characterizations of all samples were performed by using reflection high-energy electron diffraction (RHEED), field emission-scanning electron microscopy (FE-SEM), atomic force microscopy (AFM), Kelvin probe force microscopy (KPFM), Raman spectroscopy, and cathodoluminescence (CL) spectroscopy. This work also investigates the influence of substrate temperatures and Knudsen effusion cell (K-cell) temperatures of boron source on the growth of h-BN quantum dots.

## 2. Materials and Methods

### 2.1. Preparation of Substrate

SiO$_2$ film was formed on a polished and etched Si (100) wafer (Siltronix, 430 μm of thickness, and 2 inch of diameter) by an oxidation process. Then, a 100 nm-thick Ni film was deposited on SiO$_2$ by an E-gun evaporator and annealed at 800 °C for 2 min to form Ni/SiO$_2$/Si substrates.

### 2.2. The Growth of h-BN QDs

The growth of h-BN QDs on Ni/SiO$_2$/Si substrate was conducted by a ULVAC PA-MBE system. The growth chamber was equipped with high-temperature K-cell as boron sources (UMAT, slug form, 99.9999%). 0.8 sccm high-purity nitrogen (99.9999%) was used as the nitrogen source, and radio frequency power was set to 500 W. Growth conditions of h-BN QDs at different K-cell and substrate temperatures are presented in Figure 1. The growth temperature for the substrate was varied from 700, 750, and 800 °C for 2 h [26]. All substrates were subjected to a thermal cleaning at 600 °C for 30 min to remove moisture and oxide on the surface before the growth of h-BN QDs. The base working pressure was $5 \times 10^{-8}$ Pa. The K-cell temperature (T$_B$) was set to 1200 and 1300 °C, respectively. B and N were generated simultaneously during the growth. After the growth process, samples were cooled down to an ambient temperature in the chamber.

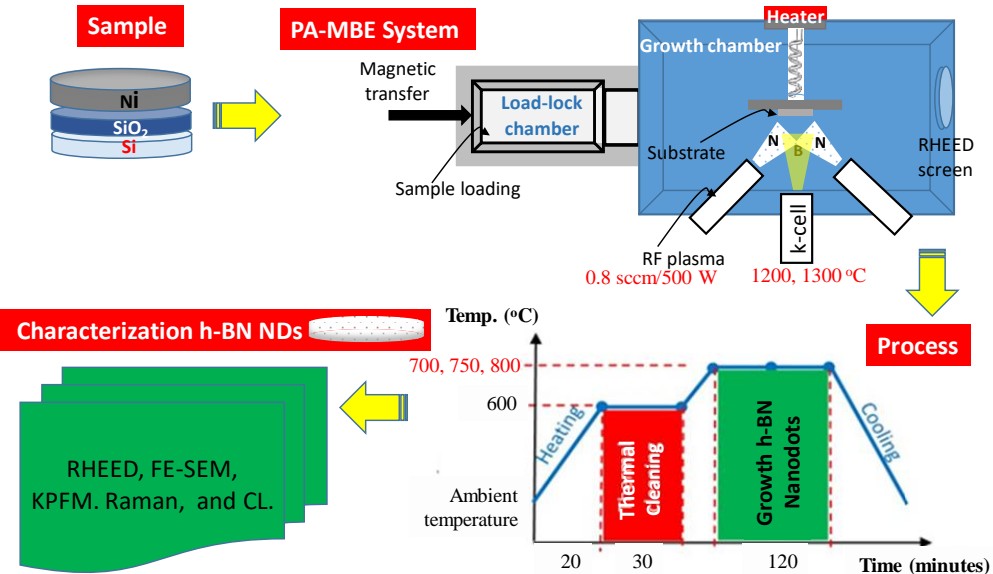

**Figure 1.** Schematic of the preparation and characterizations of h-BN QDs.

### 2.3. Characterizations of h-BN Quantum Dots

The growth process was monitored by 20 kV in situ RHEED. After the growth of h-BN QDs, the surface morphology was examined by JEOL JSM-7000F FE-SEM equipped with a silicon-draft-detector-based energy dispersive X-ray spectroscopy (EDS). AFM and KPFM (Nanosurf C3000) were employed to measure the surface roughness and the local contact voltage as work functions of materials, respectively. Raman spectroscopy (Renishaw), equipped with a 532 nm laser, was used to check the h-BN QDs. The optical property of h-BN QDs was measured by cathodoluminescence (CL) spectroscopy coupled with scanning electron microscopy.

## 3. Results and Discussion

### 3.1. Reflection High-Energy Electron Diffraction

The PA-MBE is equipped with the RHEED to determine the substrate's surface conditions and the growth of h-BN. Figure 2a is the RHEED pattern after thermal cleaning, and it looks spotty [27]. The pattern indicates that the substrate's surface quality improved as the contamination was removed [28]. When the K-cell temperatures increase from 1200 to 1300 °C with 700 °C growth temperatures in Figure 2b,c, the RHEED pattern changed from spotty to rings pattern during the growth of h-BN. Furthermore, the ring pattern was apparent when the growth temperatures increased from 700 °C to 800 °C in Figure 2c,d. The ring-type pattern indicates the polycrystalline Ni surface with h-BN.

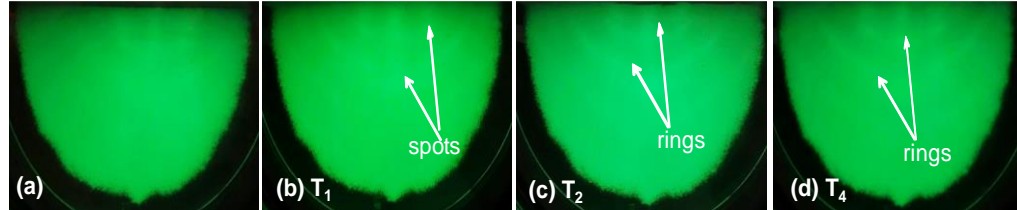

**Figure 2.** RHEED patterns: (**a**) $Ni/SiO_2/Si$ substrate after thermal cleaning, (**b**) $T_1$ (700/1200 °C), (**c**) $T_2$ (700/1300 °C), and (**d**) $T_4$ (800/1300 °C).

### 3.2. Field Emission-Scanning Electron Microscopy

FE-SEM was used to observe the surface morphology of samples. Figure 3a shows the polycrystalline Ni film with a rough surface and more grain boundary as reference. Samples $T_2$ and $T_4$ are the growth temperatures from 700 to 800 °C and K-cell temperature of 1300 °C in Figure 3b and c, respectively. Polycrystalline Ni with some h-BN nanodots as white spots was observed, and the measurement of EDS spectrum was shown in Figure 3d [29]. The formation of h-BN nanodots was preferred at the grain boundary of Ni films due to the mechanism of edge growth, which will also be supported by AFM and KPFM. The higher the growth temperature, the more h-BN nanodots can be observed. As reported [14], h-BN may have nucleated heterogeneously and the island formation in the initial stage of growth.

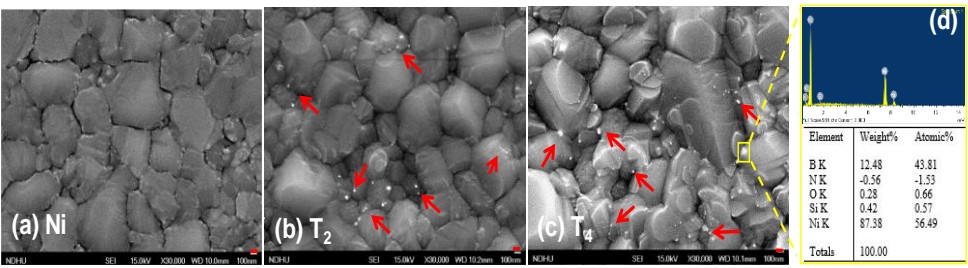

**Figure 3.** FE-SEM images, (**a**) Ni Substrate, (**b**) $T_2$ (700/1300 °C), (**c**) $T_4$ (800/1300 °C), and (**d**) EDS analysis of $T_4$.

### 3.3. Atomic and Kevin Probe Force Microscopy

For the observation of h-BN QDs, AFM and KPFM were used to investigate the surface morphology and contact potential distribution as a work function between h-BN nanodots and Ni substrate. Figure 4a shows the KPFM image with a scan area of 2 μm × 2 μm, and Figure 4c,e,g are the line scans of local contact potential difference for sample $T_2$. A random distribution of bright color spots of higher potential voltage, indicating h-BN QDs existence. At the time, the AFM image was obtained at the same region, shown in Figure 4b,d,f,h, the bright area is the top surface of the samples, and the dark area is in the valleys of the samples. In three line-scan measurements of KPFM and AFM, including a single dot, the peaks were at a different position, which also indicates the existence of the h-BN QDs.

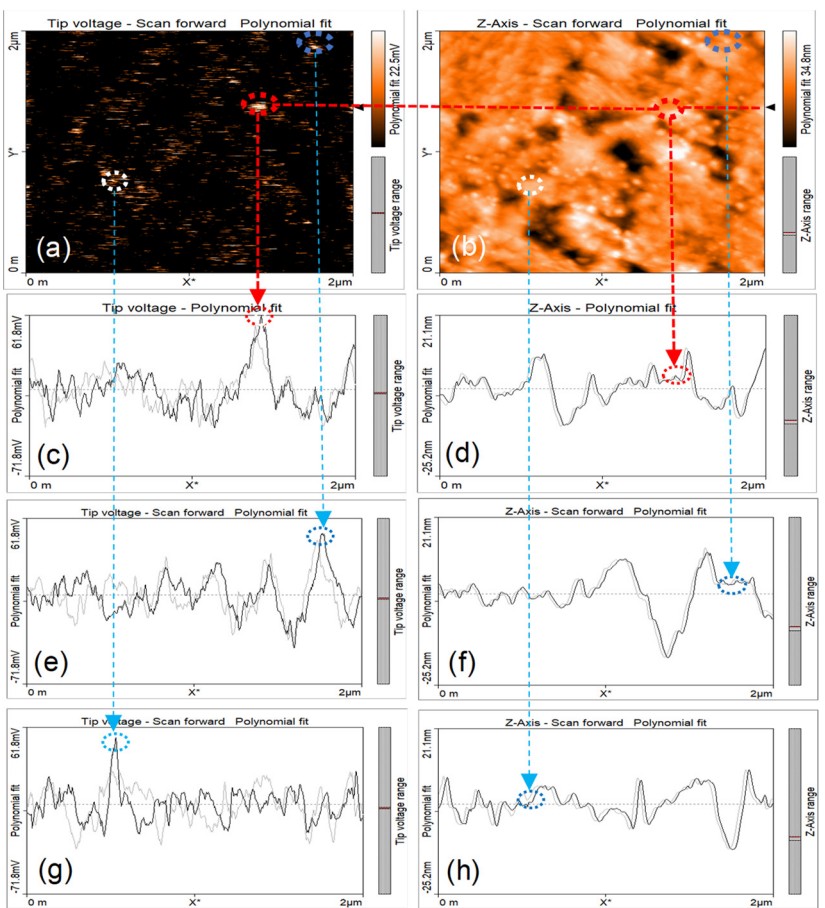

**Figure 4.** (**a,c,e,g**) KPFM images for sample $T_2$; (**b,d,f,h**) AFM images at the same region of sample $T_2$.

For different growth temperatures, the average surface roughness (Ra) of samples $T_2$, $T_3$, and $T_4$ are 4.15, 8.02 and 8.71 nm, as summarized in Table 1. The local contact potential difference between the h-BN QDs and the Ni substrate is 10.56, 9.75 and 9.63 mV, respectively. The surface roughness of Ni film increased with increasing growth temperature. Therefore, the increase in the roughness could come from the formation of h-BN QDs.

**Table 1.** Surface roughness (Ra, nm) and local contact potential difference (LCPD, mV) of $T_2$, $T_3$, and $T_4$ samples for different growth temperatures.

| Parameter | $T_2$ (700 °C) | | $T_3$ (750 °C) | | $T_4$ (800 °C) | |
|---|---|---|---|---|---|---|
| | Ra | LCPD | Ra | LCPD | Ra | LCPD |
| Average | 4.15 | 10.56 | 8.02 | 9.75 | 8.71 | 9.63 |
| Max. peak height | 20.72 | 61.82 | 29.41 | 59.19 | 47.01 | 68.13 |
| Max. peak depth | −25.23 | −64.89 | −38.63 | −55.21 | −58.04 | −62.85 |

### 3.4. Raman Analysis

Raman spectra are used to characterize the vibration mode of h-BN QDs. The Raman spectrum of the substrate was displayed in Figure 5a, and it has a similar result as the reference [28]. Then, Figure 5b–d show a weak broad peak which indicates the presence of h-BN nanodots [30]. Lorentzian fitting was used for Raman spectra. Raman shift of h-BN for $T_2$, $T_3$, and $T_4$ samples are at 1383, 1363 and 1349 cm$^{-1}$, respectively. The presence of crystalline h-BN on Ni can be confirmed by the results of Raman spectra.

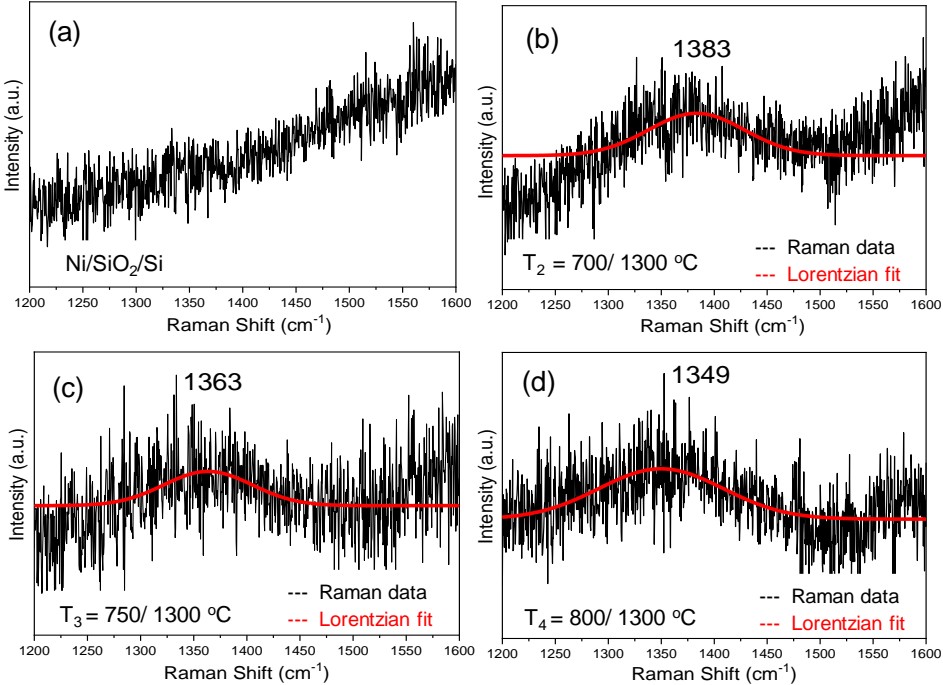

**Figure 5.** Raman spectra: (**a**) Ni/SiO2/Si substrate, (**b**) $T_2$, (**c**) $T_3$, and (**d**) $T_4$ samples.

### 3.5. Cathodoluminescence Analysis

The excitation of electrons causes the light emission process of materials. The CL emission spectra of h-BN QDs show four intense peaks at 436, 485, 546, and 610 nm for samples $T_1$ and $T_2$ in Figure 6, indicating the defect luminescence of h-BN, such as vacancies and impurities. The peak intensity increased as K-cell temperature increased to 1300 °C, which means the luminescence of h-BN QDs increased. The CL spectrum with peaks at 436 and 485 nm is attributed to the trapping centers in h-BN, as mentioned by Nistor et al. [31]. Furthermore, CL spectra for samples $T_2$ and $T_4$ in Figure 6 revealed a significant increase in the peak intensity when growth temperature increased to 800 °C, indicating the better formation of h-BN QDs. Interestingly, the peak at 436 nm disappeared as the substrate temperature increased. It could be the annealing effect to reduce the defects in h-BN QDs.

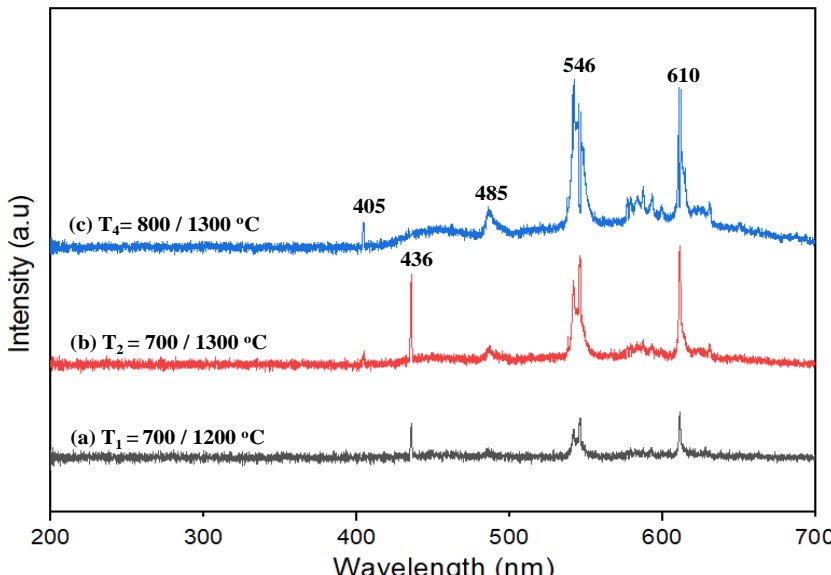

**Figure 6.** Cathodoluminescence spectra: h-BN quantum dots grown at different K-cell temperatures for samples $T_1$ (700/1200 °C) and $T_2$ (700/1300 °C), and at different substrate temperatures for samples $T_2$ (700/1300 °C) and $T_4$ (800/1300 °C).

## 4. Conclusions

In summary, the investigation of h-BN quantum dots grown on polycrystalline $Ni/SiO_2/Si$ substrates by the PA-MBE was performed. During the growth, RHEED patterns changed significantly from foggy to spotty with bright rings, indicating surface transformation to polycrystalline due to the growth of h-BN quantum dots. The results of KPFM and AFM confirm that h-BN was deposited in the form of nanodots on the substrate. Raman spectrum can perform the presence of crystal h-BN. The results of CL spectra indicated the emission of h-BN quantum dots by the defect state in the crystal. When substrate temperature and K-cell temperature of the boron source were increased, CL spectra increased in intensity and also had a sharp emission peak due to the quantum confinement effect of h-BN quantum dots.

**Author Contributions:** Conceptualization, I.-S.Y.; resources, I.-S.Y. and S.-H.C.; data curation, W.-C.H. and N.N.; writing—original draft preparation, N.N. and C.-Y.L.; writing—review and editing, I.-S.Y.; All authors have read and agreed to the published version of the manuscript.

**Funding:** This research was funded by Ministry of Science and Technology, Taiwan, grant number MOST 109-2221-E-259-004- MY3.

**Institutional Review Board Statement:** Not applicable.

**Informed Consent Statement:** Not applicable.

**Data Availability Statement:** Not applicable.

**Acknowledgments:** The authors acknowledge the Ministry of Science and Technology, Taiwan, for financially supporting this study. All authors would also like to thank Stanley Wu, Micheal Chen, and Huang-Choung Chang of ULVAC Taiwan Inc. for the support and maintaining of our PA-MBE system.

**Conflicts of Interest:** The authors declare no conflict of interest.

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
