# Peer review of "The Growth of Hexagonal Boron Nitride Quantum Dots on Polycrystalline Nickel Films by Plasma-Assisted Molecular Beam Epitaxy"

_crystals, doi:10.3390/cryst12030347_

Round 1
Reviewer 1 Report
The article is on the growth of Hexagonal Boron Nitride QDs on polycrystalline nickel films by PA-MBE. The article is well written and organized. Still the reviewer has some remarks. Please improve the RHEED image quality of Figure 2 the spots and rings are barely visible. For comparison, please add the CL spectra of the layer withouth QDs. The reviewer misses the photoluminescence spectrum at low temperatures and HR-XRD measurements as it could provide additional insight.
Author Response
Dear reviewer,
Thanks for the insightful comments to make this manuscript more complete. According to the suggestions, we made some modification and also polish the text, which was marked in red in the revised manuscript.
- RHEED images in Figure 2 have been modified in the revised manuscript. Besides, Figure 3, 4, 5 and Table 1 were modified.
- For CL spectra, we did not measure the sample without QDs.
- For the PL and HRXRD, we did it. However, the density of QDs on Ni was too low to have the observation of PL and HR-XRD.
Reviewer 2 Report
Generally, authors submitted a very well-written paper about h-BN QD fabrication. The presented manuscript refers to currently applied fabrication technologies and extends knowledge in the field of h-BN QD fabrication techniques. Only one methodological error has been detected by the reviewer:
1. Figure 3 presents EDS spectra of fabricated h-BN QD onto Ni substrate. However, no visible peaks corresponding to boron are presented on EDS spectra. Moreover, detection of boron by the EDS technique is highly unlike, due to its resolution. For boron-containing phases, the WDS (wavelength dispersive spectroscopy) should be applied. Please add the name and resolution of the used EDS detector.
Author Response
Dear reviewer,
Thanks for the appreciation of our work and providing an insightful comment to make this manuscript more complete. The model we used is JEOL JSM-7000F FE-SEM, equipped with a silicon-draft-detector based energy dispersive X-ray spectroscopy (EDS). The peak of boron in not easy to be observed by EDS, but we still got 12.48 weight% of boron in this measurement.